# Lineage-specific gene duplication and expansion of *DUF1216* gene family in *Brassicaceae*

**Zai-Bao Zhang**[1]*, **Tao Xiong**[2], **Xiao-Jia Wang**[3], **Yu-Rui Chen**[3], **Jing-Lei Wang**[3], **Cong-Li Guo**[3], **Zi-Yi Ye**[1]

**1** School of Life and Health Science, Huzhou College, Huzhou, Zhejiang, China, **2** College of Life Science, Xinyang Normal University, Xinyang, Henan, China, **3** College of International Education, Xinyang Normal University, Xinyang, Henan, China

* zaibaozhang79@163.com

**Data Availability Statement:** All relevant data are within the manuscript and its Supporting Information files.

## Abstract

Proteins containing domain of unknown function (DUF) are prevalent in eukaryotic genome. The DUF1216 proteins possess a conserved DUF1216 domain resembling to the mediator protein of Arabidopsis RNA polymerase II transcriptional subunit-like protein. The DUF1216 family are specifically existed in *Brassicaceae*, however, no comprehensive evolutionary analysis of *DUF1216* genes have been performed. We performed a first comprehensive genome-wide analysis of DUF1216 proteins in *Brassicaceae*. Totally 284 *DUF1216* genes were identified in 27 *Brassicaceae* species and classified into four subfamilies on the basis of phylogenetic analysis. The analysis of gene structure and conserved motifs revealed that *DUF1216* genes within the same subfamily exhibited similar intron/exon patterns and motif composition. The majority members of *DUF1216* genes contain a signal peptide in the N-terminal, and the ninth position of the signal peptide in most DUF1216 is cysteine. Synteny analysis revealed that segmental duplication is a major mechanism for expanding of *DUF1216* genes in *Brassica oleracea*, *Brassica juncea*, *Brassica napus*, *Lepidium meyneii*, and *Brassica carinata*, while in *Arabidopsis thaliana* and *Capsella rubella*, tandem duplication plays a major role in the expansion of the *DUF1216* gene family. The analysis of Ka/Ks (non-synonymous substitution rate/synonymous substitution rate) ratios for *DUF1216* paralogous indicated that most of gene pairs underwent purifying selection. *DUF1216* genes displayed a specifically high expression in reproductive tissues in most *Brassicaceae* species, while its expression in *Brassica juncea* was specifically high in root. Our studies offered new insights into the phylogenetic relationships, gene structures and expressional patterns of *DUF1216* members in *Brassicaceae*, which provides a foundation for future functional analysis.

## Introduction

Gene duplication provides raw materials for the production of new genes, which can be divided into tandem duplication, segmental duplication and whole genome duplication [1].

**Funding:** This work was supported by the grants from National Natural Science Foundation of China (32170351), the General Research Projects of Zhejiang Provincial Department of Education (Y202351039), Huzhou Science and Technology Plan Project (2023GZ44), and Research Program of Huzhou College (2023HXKM09). The funders had no role in study design, data collection and analysis, decision to publish, or preparation of the manuscript.

**Competing interests:** The authors have declared that no competing interests exist.

There are a wide range of different types of gene duplication in the genome [2, 3]. The eukaryotic genome contains numerous multi-copy gene families that facilitate the generation of novel protein functions [4]. Lineage-specific gene duplication and expansion refers to a notable increase in the copy numbers of certain genes in one species compared to other species, which may have played a crucial role during evolution or under environmental pressures and ultimately became fixed in that particular lineage [5–7]. Papain-like cysteine proteases in *Carica papaya* display lineage-specific gene duplication and expansion, which attributed to the ongoing reciprocal selection pressures of herbivores and plant defense mechanisms [7].

Flowers are among the most complex structures in angiosperms, playing significant roles in the evolution and advancement of sexual reproduction in plants [8]. Stamens, crucial components of flowers, are divided into anthers and filaments. Anther development in *Arabidopsis thaliana* comprises 14 stages, including anther morphogenesis, microspore development, gametophyte genesis, and anther dehiscence [9]. The tapetum, a critical component of the anther, undergoes three stages of cellular fate: tapetum cell differentiation, tapetum secretory cell formation, and tapetum cell apoptosis [9, 10]. Numerous peptides, transcription factors, kinases are expressed in the tapetum, which participate in intercellular communication and play crucial roles in plant reproductive development. Arabidopsis *TPD1* (*TAPETUM DETERMINANT 1*) encodes a cysteine-rich peptide expressed in the tapetum. Mutations in *TPD1* disrupt tapetum development, resulting in a proliferation of microsporocytes and a non-functional tapetum phenotype [11–13]. Transcription factors BES1 (BRI1 EMS SUPPRESSOR 1) and BZR1 (BRASSINAZOLE RESISTANT 1) serve as downstream targets of the TPD1-EMS1 (EXCESS MICROSPOROCYTES 1)-SERK1/2 (SOMATIC EMBRYO RECEPTOR KINASE 1/2) pathway [11] and are also expressed in the tapetum. The quintuple mutant of the *BES1* family, which consists of *bes1*, *bzr1*, *beh1*, *beh3*, and *beh4* genes, exhibits a tapetum-absent phenotype similar to that of *ems1*, *tpd1*, and *serk1serk2* mutants [11].

The *Brassicaceae* family, a part of the Brassicales order, is a large and diverse plant family with over 3709 species distributed across approximately 338 genera [14]. This family, also known as the mustard family, is predominantly found in the north temperate zone, particularly in the Mediterranean region. In China alone, there are about 95 genera and 411 species in *Brassicaceae*. *Brassicaceae* plants are characterized by unique morphological features, such as four cross-like arranged petals, which give the family its name. The flowers typically have six stamens, with two being shorter than the others, and a single pistil [15]. The fruits are often elongated capsules called siliques or shorter, rounded structures called silicles [16, 17]. Species within the *Brassicaceae* family hold significant economic and ecological importance, encompassing a wide range of crops, ornamental plants, and wild species. Notable members of this family include Brassica species, such as the model plant *Arabidopsis thaliana*, which is extensively studied for its small genome size and rapid life cycle. Other economically important species, such as *Brassica oleracea* (cabbage, broccoli, cauliflower, kale), *Brassica rapa* (turnip, Chinese cabbage), and *Brassica napus* (rapeseed, canola), are cultivated worldwide for their edible parts and oil production.

In the evolution of *Brassicaceae*, gene duplication is the main way to produce new genes [18–20], such as L-type LecRKs (Lectin Receptor Kinase) and LLPs (L-type lectin domain proteins), which makes an important contribution to their adaptation to the environment [18, 21]. Domains of unknown function (DUFs), deposited in the protein family database (Pfam), represent conserved amino acid sequences within protein domains whose functions remain uncharacterized [22, 23]. Proteins sharing the same DUF are classified into distinct DUF families [22]. The DUF protein family plays a pivotal role in the regulation of plant growth and development, mediating responses to both biotic and abiotic stresses, as well as fulfilling various regulatory functions throughout the life cycle of plants [24]. The following protein

domains have undergone functional characterization in *Arabidopsis thaliana*: DUF6 [25], DUF26 [26], DUF246 [27], DUF538 [28], DUF579 [29], DUF617 [30], DUF642 [31], DUF647 [32], DUF724 [33], DUF784 [34], DUF1117 [35], DUF1218 [36], DUF4005 [37] and DUF4228 [38]. Several of these proteins, including DUF6, DUF246, and DUF579, have been implicated in cell wall development; the involvement of the DUF538 protein has been observed in trichome development; and plant stress responses have been associated with DUF26, DUF1117, and DUF4228. In summary, these investigations have unveiled the participation of proteins containing the domain of unknown function (DUF) in a wide range of biological processes. In this study, we have identified a specific expansion of DUF1216 within the *Brassicaceae* lineage. Through comprehensive phylogenetic analysis, a total of 284 *DUF1216* genes were successfully characterized across 27 species belonging to the *Brassicaceae*, subsequently classified into four distinct subfamilies. The examination of gene structure and conserved motifs unveiled a consistent pattern in intron/exon arrangement and motif composition among *DUF1216* genes belonging to the same subfamily. Synteny analysis revealed that segmental duplication is a predominant mechanism driving the expansion of *DUF1216* genes in *B. oleracea*, *B. juncea*, *B. napus*, *L. meyneii*, and *B. carinata*. Conversely, tandem duplication plays a prominent role in the expansion of the *DUF1216* gene family in *A. thaliana* and *C. rubella*. The *DUF1216* genes exhibited a notably high expression specifically in reproductive tissues across most *Brassicaceae* species, whereas their expression was particularly elevated in the roots of *B. juncea*. Our studies have provided novel insights into the phylogenetic relationships, gene structures, and expression patterns of *DUF1216* members in *Brassicaceae*, thereby establishing a solid foundation for future functional analysis.

## Results

### Whole genome identification of *DUF1216* genes in *Brassicaceae*

Hidden Markov Model (HMM) and BLASTP search were used to identify DUF1216 proteins in 27 genome sequenced species in *Brassicaceae*. In total, 5, 7, 11, 11, 7, 6, 9, 9, 20, 18, 23, 11, 12, 11, 30, 7, 9, 5, 6, 7, 6, 14, 6, 12, 9, 7, and 6 DUF1216 proteins were retrieved from *Aethionema arabicum*, *Arabidopsis halleri*, *Arabidopsis lyrata*, *Arabidopsis thaliana*, *Arabidopsis alpina*, *Barbarea vulgaris*, *Boechera retrofracta*, *Boechera stricta*, *Brassica carinata*, *Brassica juncea*, *Brassica napus*, *Brassica nigra*, *Brassica oleracea*, *Brassica rapa*, *Camelina sativa*, *Capsella grandiflora*, *Capsella rubella*, *Cardamine hirsuta*, *Eutrema salsugineum*, *Isatis indigotica*, *Leavenworthia alabamica*, *Lepidium meyneii*, *Microthlaspi erraticum*, *Raphanus sativus*, *Schrenkiella parvula*, *Sisymbrium irio*, and *Thlaspi arvense*, respectively (Table 1 and S1 Table). No DUF1216 homologues have been found in other species, such as algae, mosses, ferns, monocotyledonous plants, dicotyledonous plants except *Brassicaceae*. Protein domain duplication is the main reason for the functional diversity of protein families [39]. About one-third (93 in 284) of the identified DUF1216 proteins contain two DUF1216 domains, indicating that these proteins are involved in complex biological processes (S1 Fig).

The physical and chemical properties of the identified DUF1216 protein were studied, including the number of amino acids, molecular weight (WM) and isoelectric point (pI) (S1 Table). The DUF1216 protein sequence varies in length from 68 to 2708 amino acids, with an average length of 602 amino acids. Its MW ranges from 7.49 to 310.10 KDa, while its pI ranges from 3.96 to 10.48.

### Phylogenetic relationship of *DUF1216* genes in *Brassicaceae*

A rootless phylogenetic tree was constructed based on the whole protein sequence of DUF1216. According to the structure of the phylogenetic tree, 284 *DUF1216* genes were

**Table 1. Distribution of *DUF1216* gene in *Brassicaceae*.**

| Species | DUF1216 Subfamily | | | | Total |
|---|---|---|---|---|---|
| | I | II | III | IV | |
| *Aethionema arabicum* | 1 | 1 | 1 | 2 | 5 |
| *Arabidopsis halleri* | 0 | 2 | 3 | 2 | 7 |
| *Arabidopsis lyrata* | 1 | 4 | 4 | 2 | 11 |
| *Arabidopsis thaliana* | 1 | 5 | 3 | 2 | 11 |
| *Arabidopsis alpina* | 1 | 2 | 2 | 2 | 7 |
| *Barbarea vulgaris* | 0 | 3 | 1 | 2 | 6 |
| *Boechera retrofracta* | 1 | 4 | 2 | 2 | 9 |
| *Boechera stricta* | 2 | 3 | 2 | 2 | 9 |
| *Brassica carinata* | 2 | 7 | 5 | 6 | 20 |
| *Brassica juncea* | 2 | 8 | 3 | 5 | 18 |
| *Brassica napus* | 3 | 9 | 5 | 6 | 23 |
| *Brassica nigra* | 2 | 5 | 2 | 2 | 11 |
| *Brassica oleracea* | 1 | 5 | 3 | 3 | 12 |
| *Brassica rapa* | 1 | 4 | 3 | 3 | 11 |
| *Camelina sativa* | 3 | 12 | 8 | 7 | 30 |
| *Capsella grandiflora* | 1 | 2 | 2 | 2 | 7 |
| *Capsella rubella* | 1 | 4 | 2 | 2 | 9 |
| *Cardamine hirsuta* | 0 | 2 | 1 | 2 | 5 |
| *Eutrema salsugineum* | 1 | 2 | 1 | 2 | 6 |
| *Isatis indigotica* | 1 | 3 | 1 | 2 | 7 |
| *Leavenworthia alabamica* | 0 | 1 | 1 | 4 | 6 |
| *Lepidium meyneii* | 3 | 3 | 2 | 6 | 14 |
| *Microthlaspi erraticum* | 1 | 1 | 2 | 2 | 6 |
| *Raphanus sativus* | 1 | 6 | 2 | 3 | 12 |
| *Schrenkiella parvula* | 2 | 3 | 2 | 2 | 9 |
| *Sisymbrium irio* | 0 | 3 | 2 | 2 | 7 |
| *Thlaspi arvense* | 1 | 1 | 2 | 2 | 6 |
| Total | 33 | 105 | 67 | 79 | 284 |

divided into four subfamilies, represented by I, II, III and IV, respectively (Fig 1). The *DUF1216* genes exhibited an uneven distribution among the four subfamilies, with 33, 105, 67 and 79 *DUF1216* genes were identified in subfamilies I, II, III and IV, respectively (Table 1). Subfamilies II, III and IV contain *DUF1216* genes from all these 27 species, whereas in subfamily I, *DUF1216* genes were lost in *A.helleri*, *B.vulgaris*, *C.hirsuta*, *L.alabamica* and *S.irio*. The *DUF1216* genes in most species are evenly distributed across four distinct subfamilies, while *C. sativa DUF1216* displayed a significant high number in subfamily II (about 40%, 12 in 30).

## Gene structure and motif conservation of DUF1216

Conservation of gene structure and motif offers crucial insights into the evolution and functional conservation of the *DUF1216* gene family. Most *DUF1216* genes (227 in 284, 80%) contain only one or two introns. Totally 51 *DUF1216* genes (51 in 284, 18%) contain more than three introns, and most of them are concentrated in subfamily I (21 in 51, 41%) (Fig 2, S1 Table). In subfamilies II, III, and IV, 84% (88/105), 67% (45/67), 76% (60/79) *DUF1216* genes have 1–2 introns, one intron, and two introns, respectively. However, in subfamily I, 64% (21/ 33) of *DUF1216* genes have over three introns.

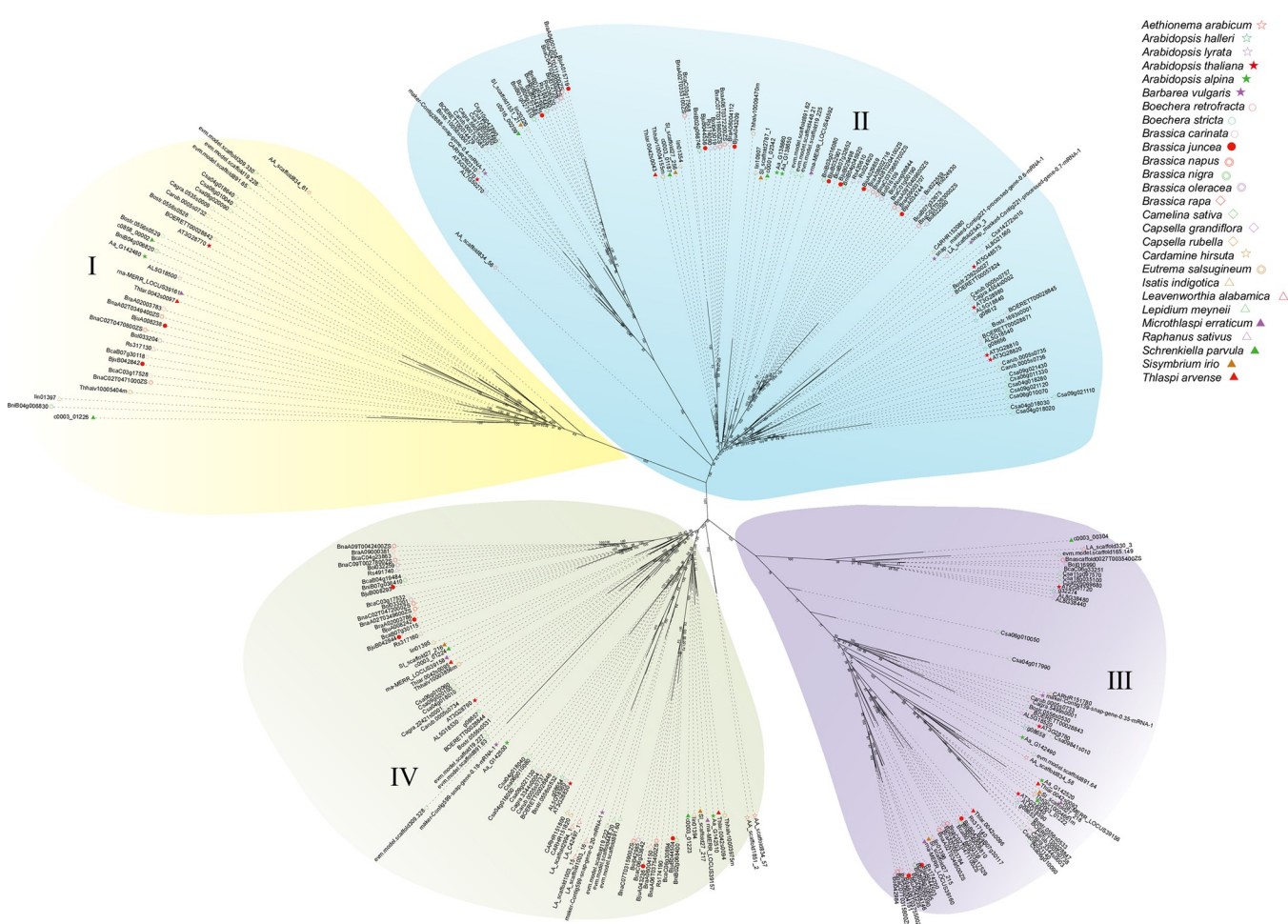

**Fig 1. Phylogenetic tree of DUF1216 in 27 species of *Brassicaceae* based on maximum likelihood method.** There are 284 DUF1216 proteins in the phylogenetic tree. These DUF1216 proteins were divided into four groups, with each branch name displayed next to the corresponding group and marked with a different color.

The conserved motif of DUF1216 protein was predicted by online software MEME, and six conserved motifs were identified (Fig 2). Motif 1 and motif 2 are widely distributed in DUF1216 protein. Motifs 4 and 5 were not identified in subfamily I with the exception of Carub.0005s0732 and Csa04g018640; motif4 was not detected in subfamily II with the exception of BjuB024425, Rs036520, Bol012128, BraA04001304, BjuA015719, and Thlar.0042s0043; motif 3 was not identified in subfamily III with the exception of AT5G61720 (Fig 2). The motif composition and position among different subfamilies are different. Motif 4 was specially identified in subfamilies III and IV. Overall, the members of DUF1216 in the same subfamily have similar motif composition, but significant differences were identified among different subfamilies, indicating the functional similarity within subfamilies and the diversity of functions between subfamilies.

## Signal peptide and subcellular localization of DUF1216 protein

Based on the predictions, most DUF1216 proteins (228 in 384, 80%) contain signal peptide, with the length of most signal peptides is approximately 25 amino acids (Fig 3, S2 Fig and S2 Table). Approximately 75.4% of the signal peptides have C (cysteine) and 14.0% of the

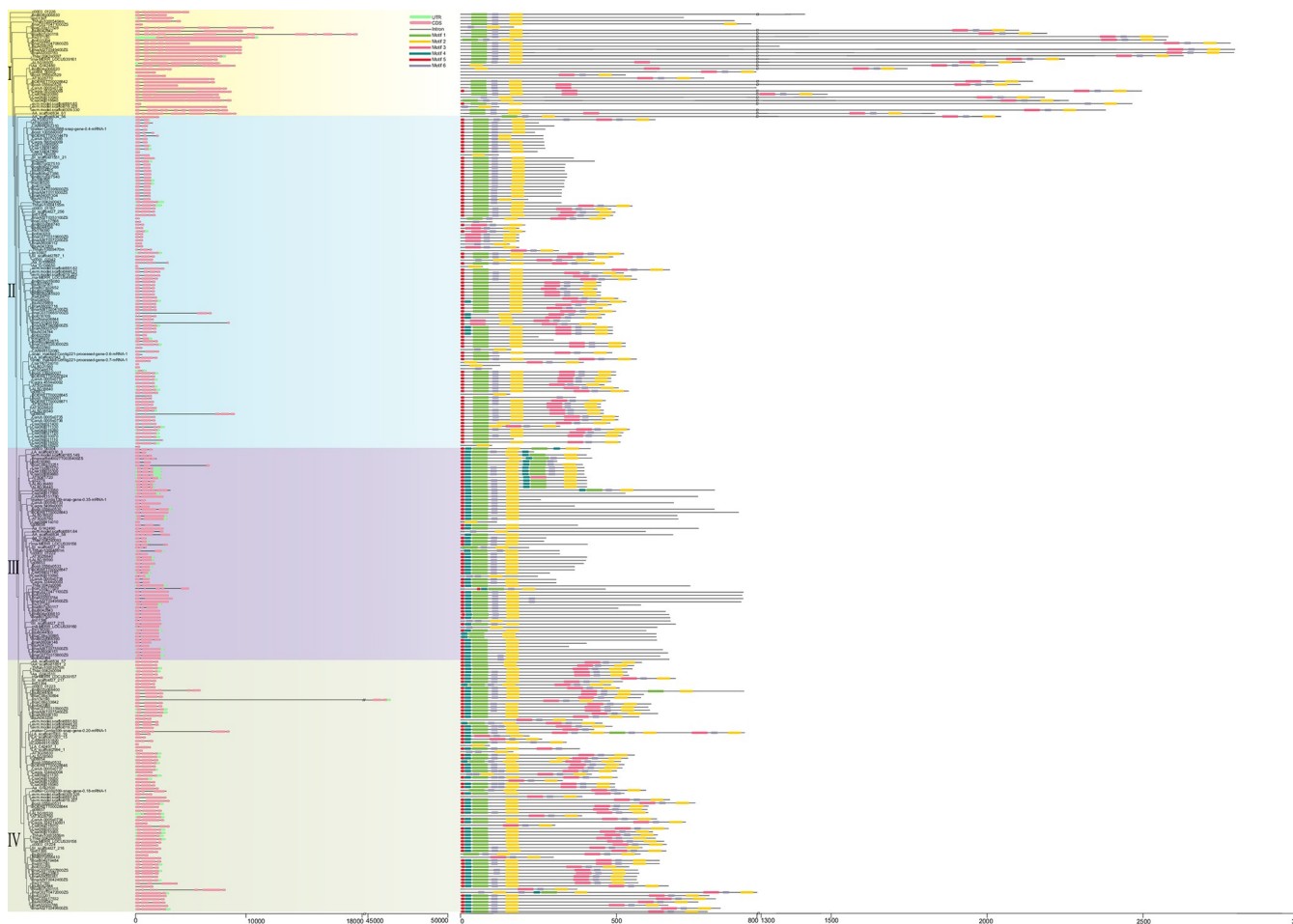

**Fig 2. Gene structure and conserved motif analysis of DUF1216 among *Brassicaceae* species.** Green boxes represent untranslated regions (UTRs), pink boxes correspond to coding sequences (CDSs), and thin black lines indicate introns. Six distinct motifs (numbered 1–6) were identified in DUF1216 proteins using the MEME program and are depicted in different colors.

signal peptides have L (leucine) at position 9, respectively. Additionally, the majority of DUF1216 members are predicted to localize to the nucleus, with only a few proteins are predicted to be located in the cell membrane, cell wall, or cytoplasm (S2 Table). These prediction results can be useful for further functional studies and as references for target screening.

## Promoter *cis*-elements analysis of *DUF1216s*

Promoter *cis*-elements play crucial roles in regulating gene expression [40]. Forty-five cis-acting elements were identified in the promoter region of *DUF1216* genes, which could be classified into four categories elements, including light-responsive, plant growth, stress, and phytohormone responsive elements (Fig 4, S3 Table). A total of 3846, 626, 2629, and 1775 light-responsive elements, plant growth elements, stress elements, and phytohormone responsive elements, were identified, respectively. In subfamily I, it is worth noting that several *cis*-acting elements, including the light-responsive elements (I-box, ATC-motif, TCT-motif, AAAC-motif) and plant growth elements (O2-site, GCN4-motif, and circadian), were absent. In subfamily II, light-responsive components (TCT-motif elements) were absent. GA-motif and AAAC-motif elements were absent in subfamily III and ACA-motif and gap-

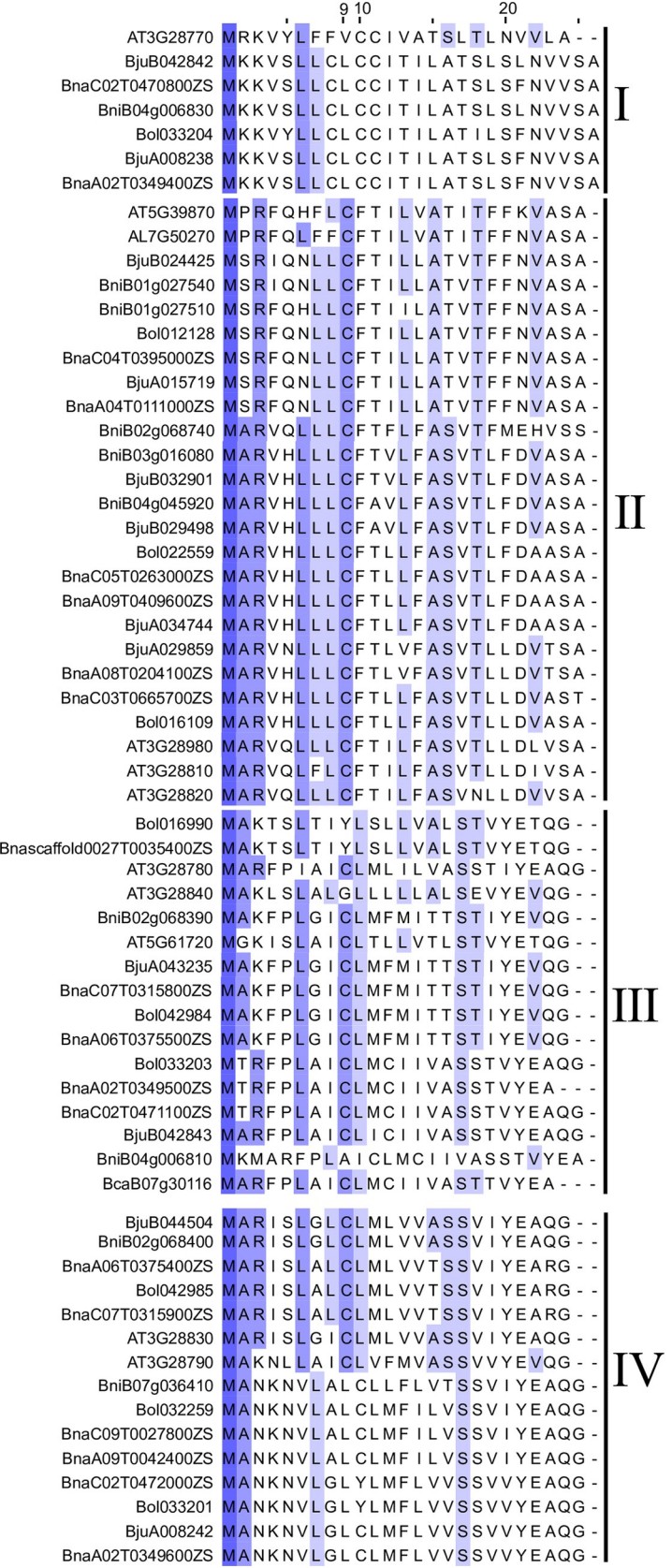

**Fig 3. DUF1216 signal peptide sequence of representative species.**

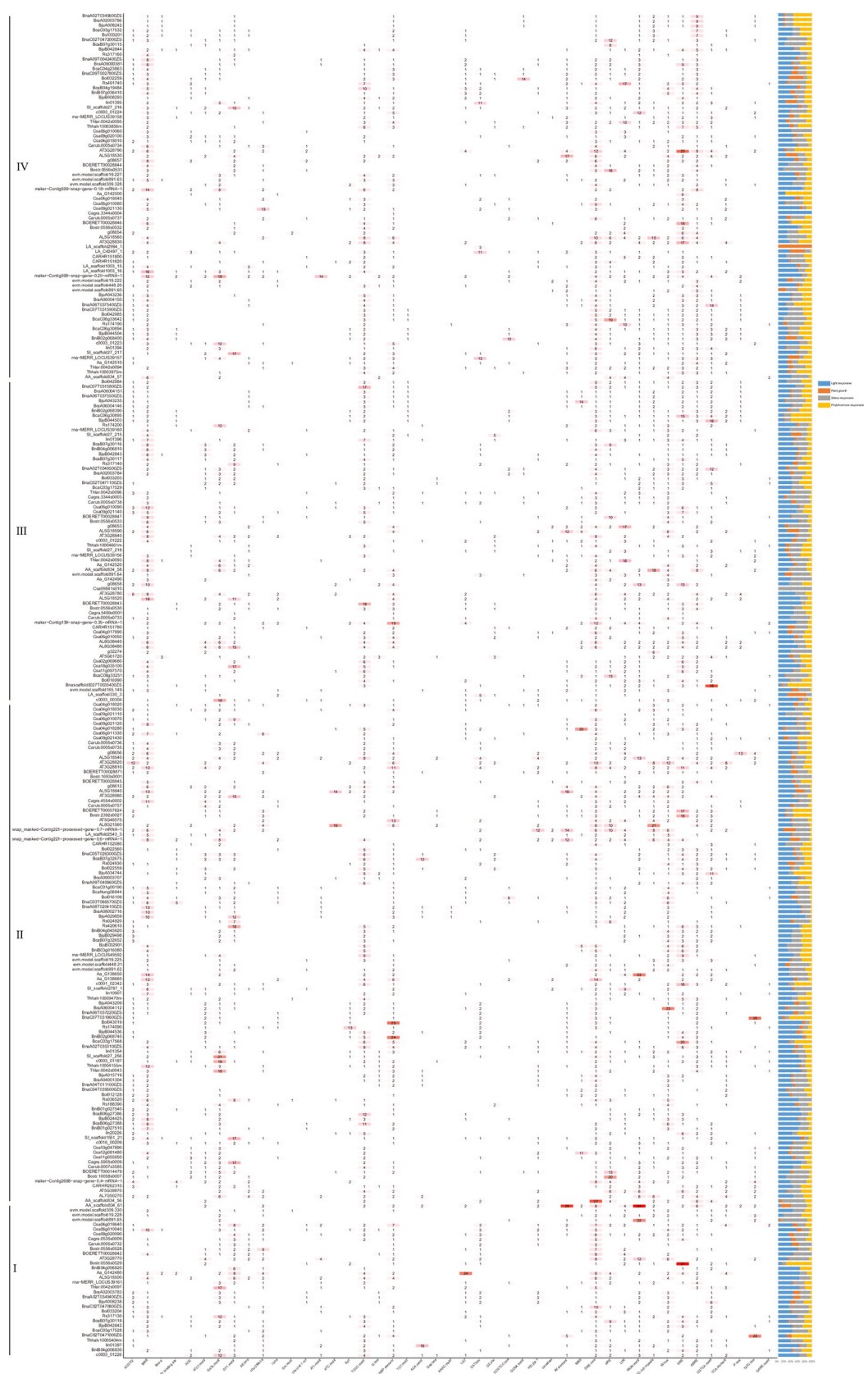

**Fig 4. DUF1216 promoter *cis*-acting element.**

box components were absent in subfamily IV. The *DUF1216* promoter are characterized by a high abundance of MRE (830) and DRE core (803) elements. These results indicate that *DUF1216s* are expressed under light induction, respond to abiotic stress, and play important roles in hormone response and regulate the plant growth.

### Gene duplication and synteny analysis of *DUF1216* genes

To explore the expansion pattern of *DUF1216* genes, the duplication pattern was analyzed (Fig 5 and S3 Fig). Previous research has indicated that segmental and tandem duplications are two primary factors contributing to the expansion of plant gene families [41, 42]. In detail, 2, 1, 3, 4, 6, 1, 1, 1, 22, 30, 12, 10 and 12 pairs of segmental duplication genes were identified in *A. thaliana*, *A.lyrata*, *S.parvula*, *B.nigra*, *B.oleracea*, *C.rubella*, *R.sativus*, *I.indigotica*, *B.juncea*, *B. napus*, *C.sativa*, *L.meyneii*, and *B.carinata*, respectively (Fig 5, S3 Fig and S4 Table). Furthermore, tandem duplication resulted in 6, 3, 3, 5, 2, 6, 2, 3, 2, 2, 16, 7, and 5 additional *DUF1216* genes in *A.thaliana*, *A.lyrata*, *S.parvula*, *B.nigra*, *B.oleracea*, *C.rubella*, *R.sativus*, *I.indigotica*, *B. juncea*, *B.napus*, *C.sativa*, *L.meyneii*, and *B.carinata*, respectively (Fig 5 and S3 Fig). Various mechanisms play significant roles in the expansion of *DUF1216* genes among different *Brassicaceae* species. Segmental duplication is the primary mechanism responsible for the expansion of *DUF1216* genes in *B.oleracea*, *B.carinata*, *B.napus*, *B.juncea*, and *L.meyneii*, while tandem duplication is the main expansion mechanism for the expansion of *DUF1216* genes in *A.thaliana* and *C.rubella*. This emphasizes the diverse evolutionary strategies utilized by distinct *Brassicaceae* species to expand *DUF1216* genes.

The analysis of Ka/Ks (non-synonymous substitution rate/synonymous substitution rate) ratios was performed to examine the selection pressure on gene evolution (S5 Table). Usually, a Ka/Ks ratio above 1 indicates positive selection, while a Ka/Ks ratio below 1 suggests purifying or negative selection [43]. These results showed that the Ka/Ks ratios of all the *DUF1216* paralogs were all less than 1, indicating that they undergone strong purifying selection pressure contributing to the maintenance of their function and reflecting that they had not diverged much during evolution. Among them, one Ka/Ks ratio was lower than 0.1 (paralog gene pair *Csa04g018280*/*Csa06g011330*) suggesting a strong purifying selection pressure and making the functions of the paralogous gene toward relative similarity.

To gain a deeper understanding of the evolutionary relationships among *DUF1216* genes, inter-species synteny was analyzed among 14 genomes, including *Arabidopsis thaliana* and *Brassica napus* (Fig 6 and S4 Fig). The synteny analysis showed that syntenic gene pairs were extensively present among these 14 genomes, such as 3 *DUF1216* syntenic gene pairs between *A.thaliana* and *A.alpina*; twelve *DUF1216* syntenic gene pairs between *A.alpina* and *B.napus*; forty-seven *DUF1216* syntenic gene pairs between *B.napus* and *C.sativa*; thirteen *DUF1216* syntenic gene pairs between *C.sativa* and *I.indigotica*; two *DUF1216* syntenic gene pairs between *I.indigotica* and *S.irio* (Fig 6). Individual homologous genes exhibited one-to-many or many-to-one homology, which was more evident in the synteny between *Arabidopsis alpina* and *Brassica napus*, *Brassica napus* and *Camelina sativa*, and *Camelina sativa* and *Isatis indigotica* (Fig 6). Furthermore, one-to-one collinear relationships were also identified between *Isatis indigotica* and *Sisymbrium irio* (Fig 6). The widespread presence of syntenic connections among these 14 genomes suggests that whole genome duplication significantly contributed to the expansion of *DUF1216* families.

### Expression profile of *DUF1216* genes

The expression of *DUF1216* genes in different tissues was detected by using public RNA-seq data (Fig 7). In *Arabidopsis thaliana*, 23 tissues and developmental stages were investigated

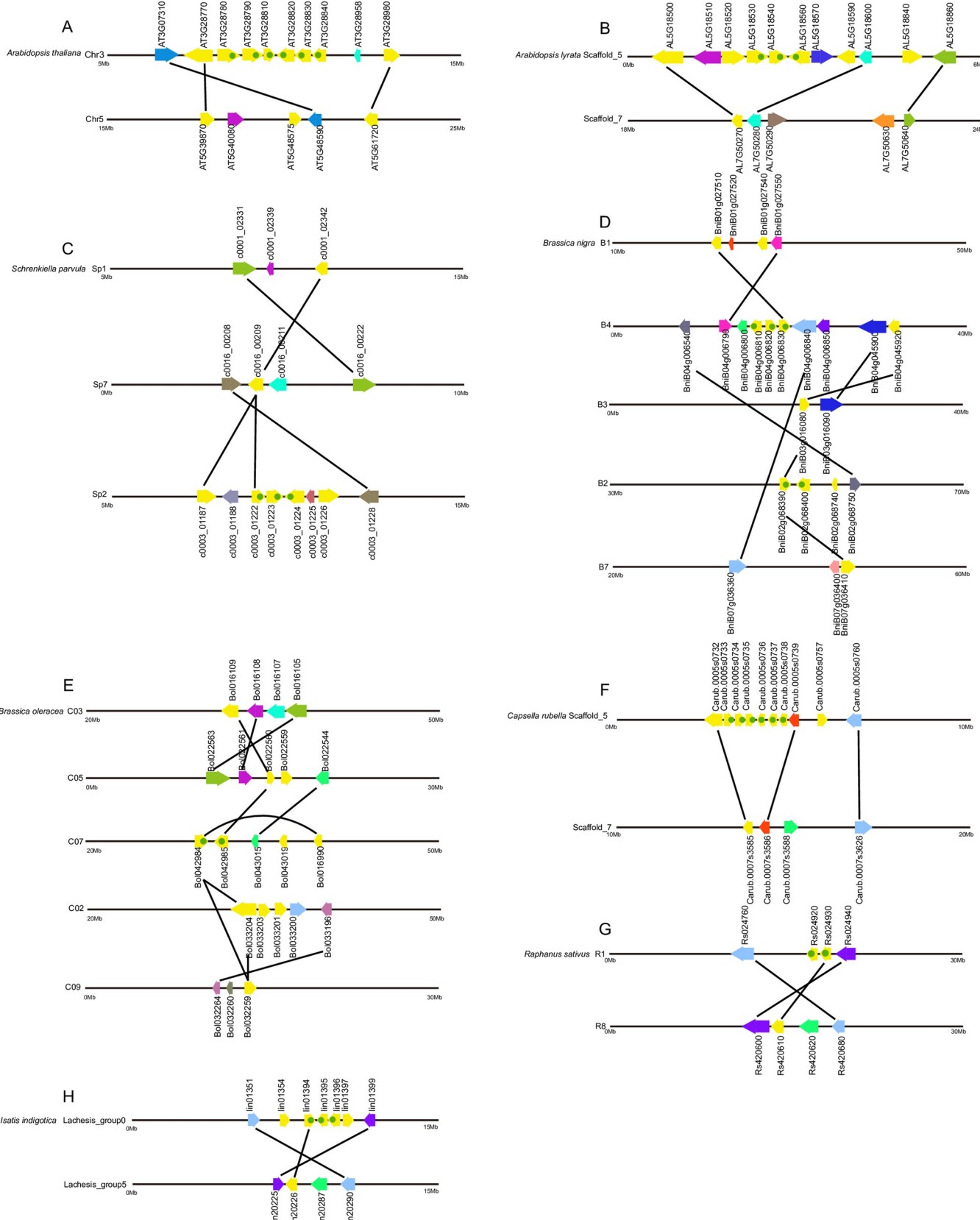

**Fig 5. Intraspecies syntenic relationships of DUF1216 genes in various plant species.** A, *Arabidopsis thaliana*; B, *Arabidopsis lyrata*; C, *Schrenkiella parvula*; D, *Brassica nigra*; E, *Brassica oleracea*; F, *Capsella rubella*; G, *Raphanus sativus*; H, *Isatis indigotica*. Yellow highlights represent *DUF1216* genes; tandem duplications denoted by yellow arrows with green dots; syntenic genes linked by black lines.

(Fig 7A). Ten *AtDUF1216* genes were significantly high expression in mature pollen and stamen of stage 15 flowers, while its transcripts in vegetable organs were barely detected. These results indicate that Arabidopsis *DUF1216* genes function in stamen development. In *Arabidopsis lyrata*, 7 tissues were examined, i.e., leaf, mature stigma, mature pollen, bolting flower, mature stem, mature root, and young silique (Fig 7B). The *A.lyrata DUF1216* genes also exhibit high expression levels in mature pollen and bolting flower, similar to that in *Arabidopsis thaliana*. In *Arabidopsis halleri*, five tissues were investigated including old leaves, new leaves, roots, self-pollinated pistils, and unpollinated pistils (Fig 7C). All *A.halleri DUF1216* genes exhibited high expression in both self-pollinated and unpollinated pistils, mirroring the expression patterns observed in Arabidopsis and *Arabidopsis lyrata*. Most *DUF1216* genes in *Brassica napus*, *Camelina sativa*, *Brassica oleracea*, *Brassica carinata*, *Brassica nigra*, *Brassica carinata*, *Brassica oleracea*, and *Brassica rapa*, displayed high expression in reproductive tissues, especially in the male organs (Fig 7E–7J). These expression patterns were consistent with the Arabidopsis *DUF1216* genes. In *Brassica juncea*, the *DUF1216* genes showed different expression patterns (Fig 7D), with most *DUF1216* genes were highly expressed in roots, indicating the special functions of *B.juncea DUF1216* in root development.

The majority of *AtDUF1216* genes exhibit downregulation in anther mutants, such as *DYT1* (*DYSFUNCTIONAL TAPETUM1*), *TDF1* (*TAPETAL DEVELOPMENT and FUNCTION1*) (Fig 7K), *AMS* (*ABORTED MICROSPORES*), and *MS188* (*MALE STERILITY188*). This suggests that the expression of *AtDUF1216* genes is under the regulation of anther-specific transcription factors.

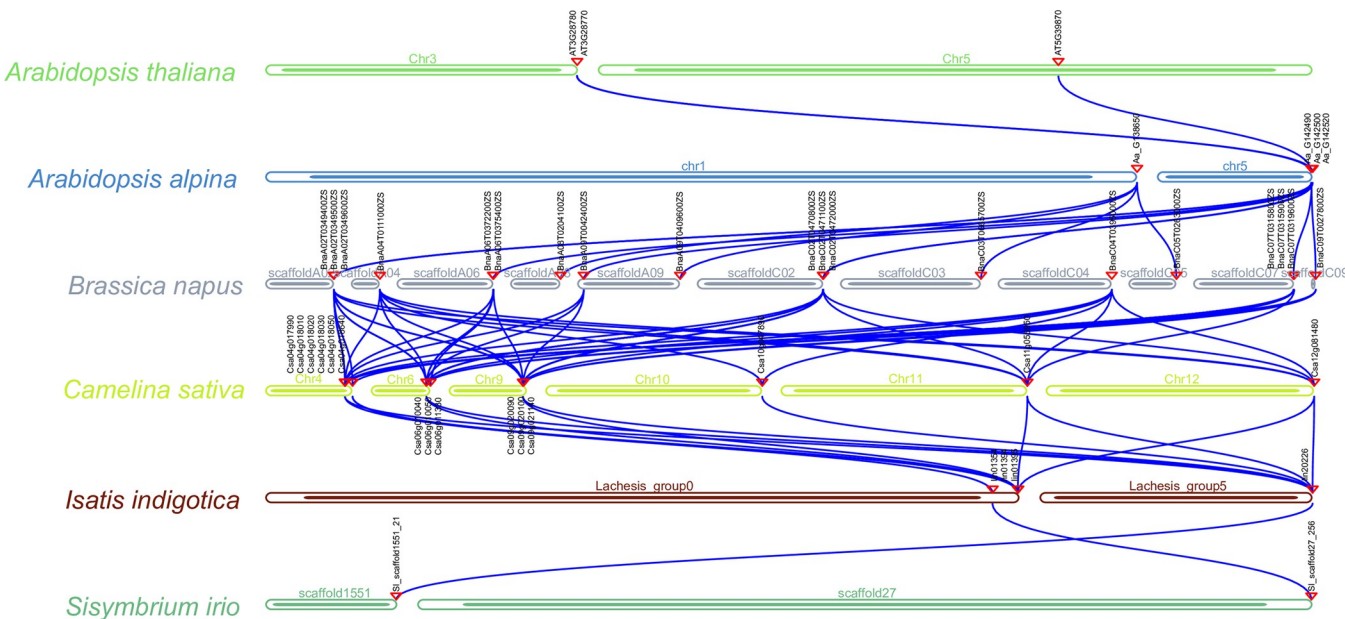

**Fig 6. Interspecies syntenic relationship of *DUF1216* genes in *Brassicaceae*.** Orthologous gene synteny represented by black lines.

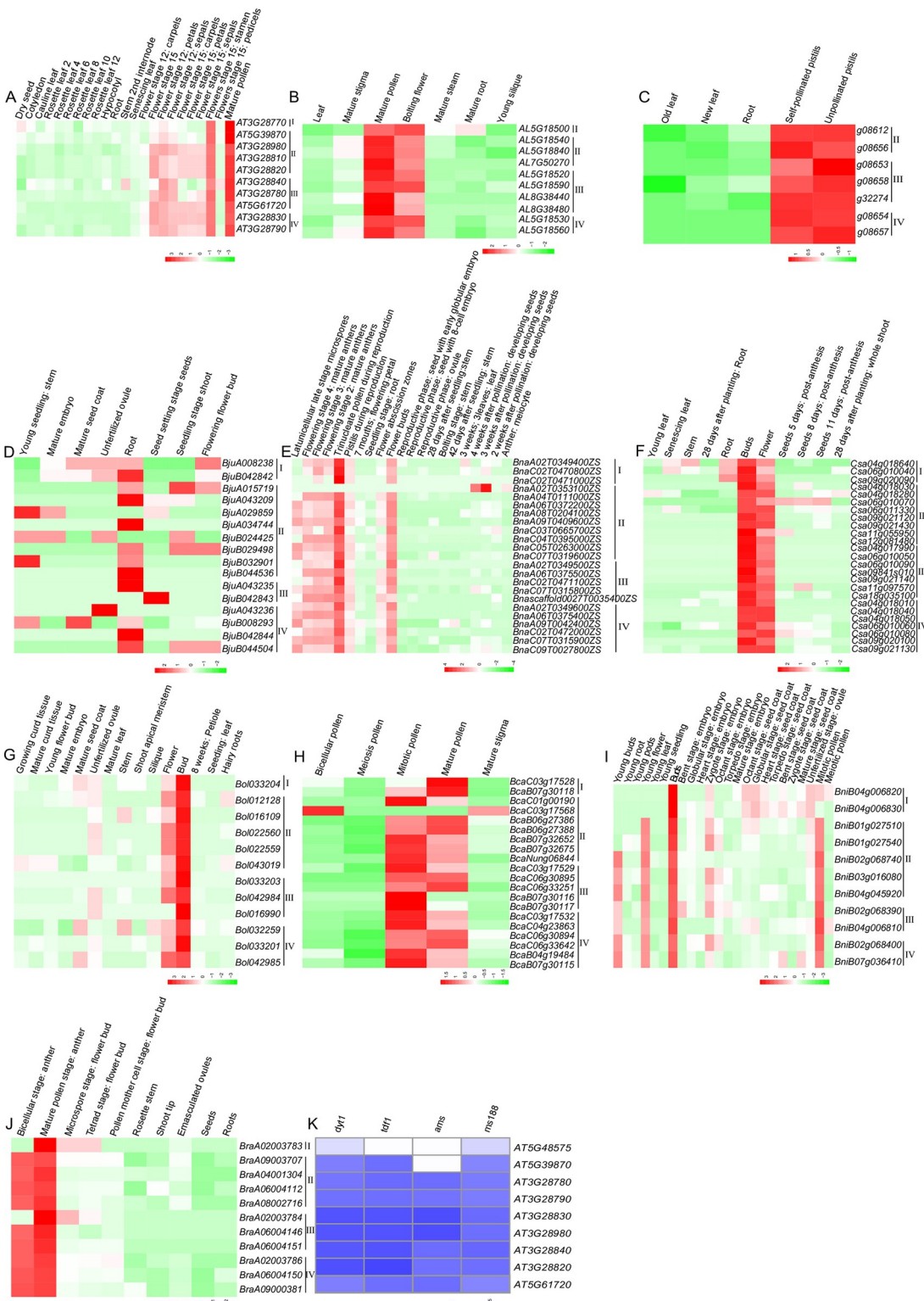

**Fig 7. Expression patterns of *DUF1216* genes in various plant tissues.** A, *Arabidopsis thaliana*; B, *Arabidopsis lyrata*; C, *Arabidopsis halleri*; D, *Brassica juncea*; E, *Brassica napus*; F, *Camelina sativa*; G, *Brassica oleracea*; H, *Brassica carinata*; I, *Brassica nigra*; J, *Brassica rapa*; K, Anther mutants in *Arabidopsis thaliana*.

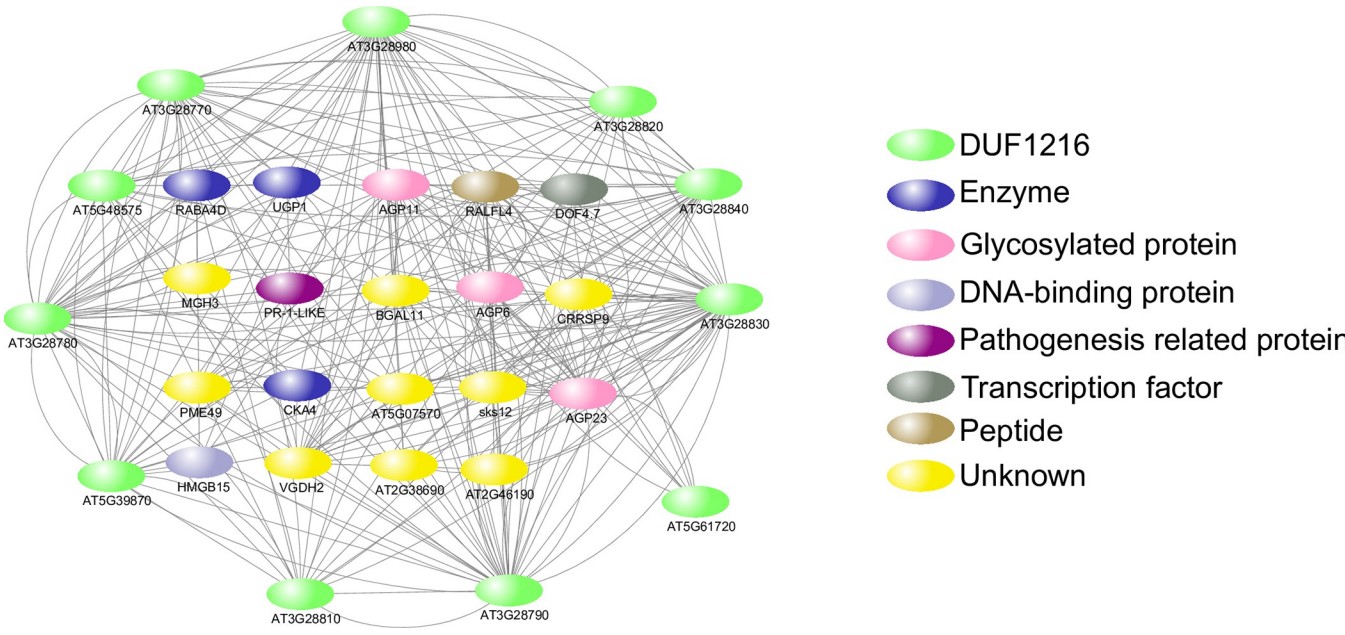

**Fig 8. Co-expression network of *DUF1216* genes in *Arabidopsis thaliana*.**

## Analysis of *DUF1216* genes co-expression network

To analyze the interaction of *DUF1216* genes with other genes, co-expression analysis was performed for Arabidopsis *DUF1216* genes (Fig 8). The 11 Arabidopsis *DUF1216* genes are co-expressed with each other and are interacting with a total of 19 transcripts. Many of these co-expressed transcripts were found to function in pollen development. For example, *AtUGP1* (*UDP-glucose pyrophosphorylase 1*) and *AtUGP2* (*UDP-glucose pyrophosphorylase 2*) encode UGPase, express in all organs, and the *ugp1 ugp2* double mutant cause male sterility [44]. *AGP6* (*Arabinogalactan protein 6*) and *AGP11* (*Arabinogalactan protein 11*), encode cell wall proteoglycans, expressed in pollen, and mutation in *AGP6* and *AGP11* impaired pollen development and hindered pollen tube growth [45–48]. *AGP23* is expressed exclusively in pollen and may play a significant role in microspore development and/or pollen tube growth [49]. *RALF4* (*RAPID ALKALIZATION FACTOR*) is expressed in pollen tube and inhibits pollen germination. [50]. *AtHMGB15* (*ARID-HMG DNA-binding protein 15*) is predominantly expressed in pollen grains and pollen tubes, and mutation in *AtHMGB15* result in impaired pollen tube growth and reduced seed set rates [51]. *RABA4D*, a Rab GTPase gene, is specifically expressed in pollen tube and regulates both PRK6 distribution and pollen tube growth [52–54]. In summary, *AtDUF1216* genes are co-expressed with various genes crucial for pollen development, suggesting their potential importance in these processes.

## Gene expression analysis qRT-PCR validation

To verify the expression profiles of *DUF1216* genes, we selected and analyzed the expression of Arabidopsis *DUF1216* genes with qRT-PCR (Fig 9). The expression levels of the tested Arabidopsis *DUF1216* genes were specifically high expressed in mature flowers and pollen grains, but not in vegetable tissues (leaf, root, stem) (Fig 9A). These expression patterns are similar to transcriptomic results (Fig 7), indicating that Arabidopsis *DUF1216* genes involved in reproductive development.

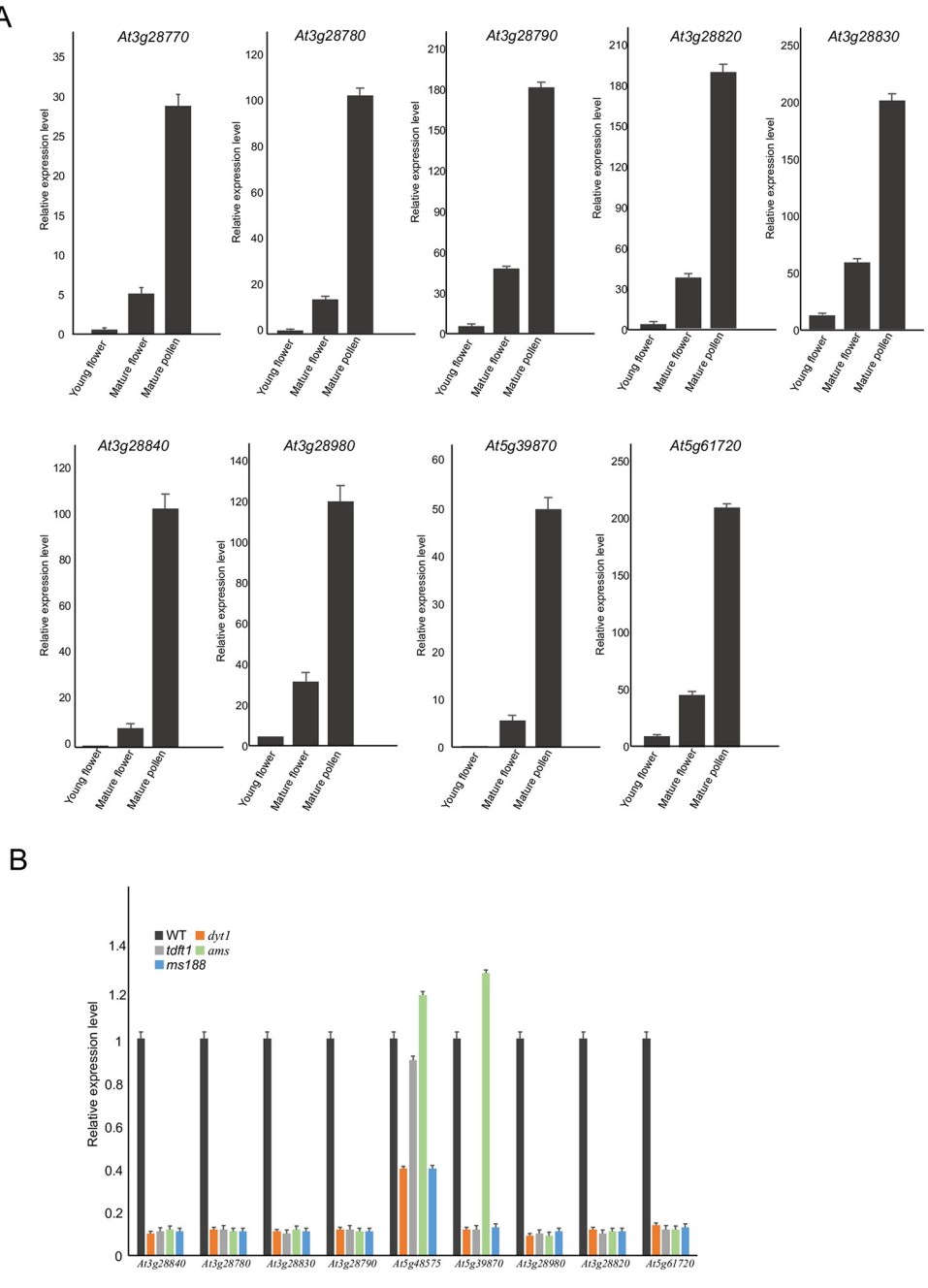

**Fig 9. Expression profiles of Arabidopsis *DUF1216* genes.** A, Expression profiles of Arabidopsis *DUF1216* genes in young flower, mature flower and mature pollen. The error bars were obtained from three measurements. B, qRT-PCR analysis of *DUF1216* genes in the inflorescence of WT, *dyt1*, *tdf1*, *ams*, and *ms188* mutants. Data are presented as mean and SD (n = 3).

Dyt1, TDF1, AMS and MS188 are essential transcription factors that regulate anther development [55–59]. Based on the previous microarray data, most *DUF1216* genes act downstream of these regulators (Fig 7) [59]. We collected the inflorescences of *dyt1*, *tdf1*, *ams*, and *ms188*, for quantitative real-time PCR (qPCR) analysis. The qPCR results showed that the expressions of *DUF1216* genes were downregulated in these mutants (Fig 9B). These results demonstrated

that Arabidopsis *DUF1216* genes were regulated by anther important transcription factors during anther development.

## Discussion

The Brassicaceae family holds significant scientific and economic importance, particularly in the fields of agriculture and human health [60–62]. The *DUF1216* are existed exclusively in *Brassicaceae*. However, it is unclear how *DUF1216* family has evolved in *Brassicaceae* plants. Here, we conducted a comprehensive evolutionary analysis of *DUF1216* genes in *Brassicaceae*. Our studies provided new insights into the phylogenetic relationships, gene structures, and expression patterns of *DUF1216* genes in *Brassicaceae*. These findings lay the foundation for future functional analysis.

This study represents the first comparative evolutionary analysis of the *DUF1216* genes across *Brassicaceae* species. The number of *DUF1216* genes exhibited significant differences among various species of *Brassicaceae*. Most *Brassicaceae* plants have approximately 10 *DUF1216* genes, while *Brassica carinata*, *Brassica juncea*, *Brassica napus*, and *Camelina sativa* exhibit a higher number of *DUF1216* genes at 20, 18, 23, and 30, respectively (Table 1). Thus, even though the genome sizes of different *Brassicaceae* species are similar, the copy number of the *DUF1216* gene varies among them.

Phylogenetic analysis indicates that the *DUF1216* genes in *Brassicaceae* can be classified into four subfamilies (Fig 1). All DUF1216 proteins, except for AT5G61720, contain the conserved DUF1216 domain. Additionally, around one-third of DUF1216 proteins contain two DUF1216 domains. These results suggest that the DUF1216 domain was lost in AT5G61720 during evolution, and that the protein with two DUF1216 domains may have some additional functions. The majority of *DUF1216* genes in subfamily I contained more than three introns, whereas most genes in the other three subfamilies contained 1–2 introns (Fig 2). These results indicating that the subfamily I *DUF1216* genes may have distinct functions in comparison to those in the other subfamilies. The arrangement of motif within the subfamily was generally consistent, but there were significant differences among different subfamilies (Fig 2). Motif4 is exclusively existed in subfamilies III and IV, and Motif3 is exclusively present in subfamilies I, II, and IV. The existence of different motifs among different DUF1216 proteins may endow different functions, resulting in functional differentiation.

Gene duplications play crucial roles in the emergence of new gene functions and the expansion of gene families [63]. Our study emphasizes the diverse evolutionary strategies utilized by distinct *Brassicaceae* species to expand the *DUF1216* genes (Fig 5, S3 Fig and S4 Table). The exclusive expansion of *DUF1216* genes in *Brassicaceae* plants suggests that these genes are biologically significant within this specific plant family. In addition, the Ka/Ks values of the duplicated *DUF1216* genes are less than 1 (S5 Table), indicating that they have undergone a more intensive purification selection. Extensive interspecific synteny among *DUF1216* genes was identified, indicating that *DUF1216* are conserved during evolution in *Brassicaceae* (Fig 6 and S4 Fig).

Most *DUF1216* genes displayed high expression in reproductive organs, indicating that they involved in reproductive development (Fig 7). However, in *Brassica juncea*, the *DUF1216* genes are highly expressed in roots, indicating a different role of *B.juncea DUF1216* genes in root growth. In Arabidopsis, most of the *DUF1216* genes were located downstream of anther transcription factors, and *AtDUF1216* genes were co-expressing with pollen-related genes. These results suggest that *AtDUF1216* genes may play important regulatory roles in pollen development.

## Conclusions

In this study, we conducted a comprehensive and genome-wide investigation of the *DUF1216* gene family in representative *Brassicaceae* species for the first time. We analyzed gene structure, protein motif, synteny relationship, expression, and phylogenomic relationship of *DUF1216* genes, thereby providing novel insights into the evolution of this gene family. Our research offers valuable information for future functional analysis and utilization of *DUF1216* genes in *Brassicaceae*.

## Materials and methods

### Data sources and sequence acquisition

Arabidopsis *DUF1216* genes were retrieved from the Arabidopsis Information Resource (TAIR, https://www.arabidopsis.org/). Genomic sequences of *Brassicaceae* plants, *Aethionema arabicum*, *Arabidopsis halleri*, *Arabidopsis lyrata*, *Arabidopsis alpina*, *Barbarea vulgaris*, *Boechera retrofracta*, *Boechera stricta*, *Brassica carinata*, *Brassica juncea*, *Brassica napus*, *Brassica nigra*, *Brassica oleracea*, *Brassica rapa*, *Camelina sativa*, *Capsella grandiflora*, *Capsella rubella*, *Cardamine hirsute*, *Eutrema salsugineum*, *Isatis indigotica*, *Leavenworthia alabamica*, *Lepidium meyneii*, *Microthlaspi erraticum*, *Raphanus sativus*, *Schrenkiella parvula*, *Sisymbrium irio* and *Thlaspi arvense*, were downloaded from TBGR (http://www.tbgr.org.cn/download/download.html) [64–87].

Two methods were employed to identify *DUF1216* genes in *Brassicaceae*. First, HMMER search (E-value = 1e−10) with the Hidden Markov Model profile of the DUF1216 domain (PF06746) was conducted to search local databases [88, 89]. Second, the amino acid sequences of *Arabidopsis thaliana* DUF1216 members were used to perform a BLASTP search against the protein database with an E-value less than $10^{-6}$ [90]. The putative *DUF1216* genes were further validated using online tools such as CDD (https://www.ncbi.nlm.nih.gov/Structure/cdd/wrpsb.cgi/) [91], HMM (https://www.ebi.ac.uk/interpro/) [92] and SMART (https://smart.embl-heidelberg.de/) [93].

### Multiple sequence alignment, protein structure predictions and phylogenetic analysis

DUF1216 multiple sequence alignments were performed using MAFFT software [94]. Phylogenetic trees were generated based on full protein sequences. The maximum likelihood (ML) phylogenetic tree was constructed using IQ-TREE with the parameter '-m MFP -bb/alrt 1000' and 1000 ultra-bootstrap replicates [95]. Branch support was tested using 1000 bootstrap replicates. The consensus trees topology was visualized with iTOL [96].

### Gene structure, conserved motif analysis, signal peptide prediction and subcellular localization prediction

The protein properties were analyzed using ProtParam (http://web.expasy.org/protparam/), while the gene structures were visualized using the Gene Structure Display Server (GSDS) (http://gsds.cbi.pku.edu.cn/index.php). Conserved motifs were predicted utilizingMEME (http://meme.nbcr.net/meme3/mme.html) [97]. The signal peptides were identified using SignalP (https://services.healthtech.dtu.dk/services/SignalP-5.0/) [98], while subcellular localization was predicted utilizing Plant-mPLoc (http://www.csbio.sjtu.edu.cn/bioinf/plant-multi/) [99].

## Promoter *cis*-acting element and synteny analysis

The analysis of promoter cis-elements was performed using the Plant Care website (http://bioinformatics.psb.ugent.be/webtools/plantcare/html/) [100], while R software was utilized for data visualization. The chromosomal positions of *DUF1216* genes were determined based on comprehensive genome annotation. The gene duplication patterns were identified using MCScanX with default settings [101], while the synteny map was generated using CIRCOS to visually connect putative duplicated genes.

## Gene expression and co-expression network analysis

The transcriptional levels of *DUF1216* genes assessed by analyzing transcriptome datasets obtained from the National Center for Biotechnology Information (NCBI) database (https://www.ncbi.nlm.nih.gov/). The reads were subjected to reanalysis, and the quantification of gene expression was performed as fragments per kilobase per million reads (FPKM).The heatmap with k-means clustering was generated using the R software. The mRNA expression data for four mutants (*dyt1*, *tdf1*, *ams*, and *ms188*) were obtained from a previous study [102]. The co-expression network was evaluated using the Arabidopsis Information Resource (TAIR, https://www.arabidopsis.org/) database, while Cytoscape software was employed for visualization.

## RNA extraction and quantitative reverse transcription-PCR

The total RNA was isolated using TRIzol reagent (Invitrogen, Thermo Fisher Scientific, United States) according to the instructions of the manufacturer. The first-strand cDNAs were synthesized from DNase I-treated total RNA using the cDNA Synthesis Kit (Takara, Japan). The gene expression was normalized using tubulin beta8 (TUB8) (At5g23860) as the reference gene, and the relative gene expression was calculated as the mean of three biological replicates and three technical replicates. Relative expression was calculated by the $2^{-\Delta Ct}$ and $2^{-\Delta\Delta Ct}$ methods. Gene specific primers used for quantitative reverse transcription-PCR (qRT-PCR) are listed in S6 Table.

## Supporting information

**S1 Fig. DUF1216 protein sequence.** DUF1216 domains are framed with red lines.
(PDF)

**S2 Fig. DUF1216 signal peptide sequence.**
(PDF)

**S3 Fig. The intraspecies syntenic relationship of *DUF1216* genes in *Brassica juncea*, *Brassica napus*, *Camelina sativa*, *Lepidium meyneii*, *Brassica juncea* and *Brassica carinata*.** Synteny genes are linked by black lines.
(PDF)

**S4 Fig. Synteny analysis of *DUF1216* among 14 species of *Brassicaceae*.**
(PDF)

**S1 Table. *DUF1216* family genes identified in 27 *Brassicaceae* plants.**
(XLSX)

**S2 Table. DUF1216 signal peptide and subcellular localization prediction.**
(XLSX)

**S3 Table. *DUF1216* promoter cis-acting element data.**
(XLSX)

**S4 Table. Segmental duplication gene pairs.**
(XLSX)

**S5 Table. The Ka/Ks analysis of DUF1216 genes.**
(XLSX)

**S6 Table. Primers used in qPCR.**
(XLSX)

**S7 Table. DUF1216 protein sequences used to build phylogenetic tree.**
(DOCX)

## Author Contributions

**Funding acquisition:** Zai-Bao Zhang.

**Visualization:** Xiao-Jia Wang, Yu-Rui Chen, Jing-Lei Wang, Cong-Li Guo, Zi-Yi Ye.

**Writing – original draft:** Zai-Bao Zhang, Tao Xiong.

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
