## [Decision Letter · Decision Letter 0]

5 Mar 2024

PONE-D-24-02626Lineage-specific gene duplication and expansion of DUF1216 gene family in BrassicaceaePLOS ONE

Dear Dr. Zhang,

Thank you for submitting your manuscript to PLOS ONE. After careful consideration, we feel that it has merit but does not fully meet PLOS ONE’s publication criteria as it currently stands. Therefore, we invite you to submit a revised version of the manuscript that addresses the points raised during the review process.

I agree with all the reviewers who suggested that experimental validation would be helpful. You can use qRT-PCR to validate some of your results.

I have verified that the reviewer's name is not the author of any of the papers they have suggested for citations. Please make sure they are relevant if you want to add any of them.

The paper needs to be proofread thoroughly.

We look forward to receiving your revised manuscript.

Kind regards,

Achraf El Allali, PhD

Academic Editor

PLOS ONE

“National Natural Science Foundation of China (32170351)”

“This work was supported by the grants from National Natural Science Foundation of China (32170351). The authors declare that they have no competing interests.”

“National Natural Science Foundation of China (32170351)”

5. We note that your Data Availability Statement is currently as follows: [The data underlying the results presented in the study are available from (include the name of the third party and contact information or URL).]

Additional Editor Comments:

I agree with all the reviewers who suggested that experimental validation would be helpful. You can use qRT-PCR to validate some of your results.

I have verified that the reviewer's name is not the author of any of the papers they have suggested for citations. Please make sure they are relevant if you want to add any of them.

The paper needs to be proofread thoroughly.

Reviewers' comments:

Reviewer's Responses to Questions

**Comments to the Author**

1. Is the manuscript technically sound, and do the data support the conclusions?

Reviewer #1: Partly

Reviewer #2: Yes

Reviewer #3: Yes

2. Has the statistical analysis been performed appropriately and rigorously? 

Reviewer #1: Yes

Reviewer #2: Yes

Reviewer #3: Yes

3. Have the authors made all data underlying the findings in their manuscript fully available?

Reviewer #1: Yes

Reviewer #2: Yes

Reviewer #3: Yes

4. Is the manuscript presented in an intelligible fashion and written in standard English?

Reviewer #1: No

Reviewer #2: Yes

Reviewer #3: No

5. Review Comments to the Author

Reviewer #1: I suggest authors to check the English by a native English speaker. More comments have been upload in the online system.

• I suggest authors to check the English by a native English speaker

• Italicize the word “Brassicaceae” througout the manuscript

• Avoid starting senetence with a mathematical figure (see line 36)

• Check for space (see lin3 36)

• All scientific names should be italicized for an example see line 74.

• The section of introduction is underdeveloped. Also, it misses out on several important references like authors gene duplication is the main way to produce new gene in Brassica, such statements need to be cited. The authors must mention about other gene families (at least two) which experienced gene duplication events.

• How do DUF1216 modulate gene duplication?

• To analyze phylogeny, on what basis different listed plants were selected.

• In the methodology section, references are not cited for some context for the protocols used for experimentation by the author. Please cite appropriate references along with the text to acknowledge the contribution of other researchers and to justify the authenticity of the protocols used. For example, cite for HMMER, IQ-TREE.

• There is tense inconsistency in methodology. Kindly check and rectify as it a grammatical error which must be rectified for uniform reading of manuscript.

• Authors must add more clarity about the Ka/Ks related results obtained. What can be deduced if a gene family experience purifying selection pressure.

• It is suggested that qRT-PCR will add more meaning in your expression experiment work.

• What is the contribution of this research in field of science and to the society?

You can add citation from these articles just to add general importance of brassica.

• Farooq, O., Ali, M., Sarwar, N., Mazhar Iqbal, M., Naz, T., Asghar, M., & Ehsan, F. (2021). Foliar applied brassica water extract improves the seedling development of wheat and chickpea. Asian Journal of Agriculture and Biology, 1.

• Ahmad, N., Fazli Ahad, R., Iqbal, T., Khan, N., Nauman, M., & Hameed, F. (2020). Genetic analysis of biochemical traits in F3 populations of rapeseed (Brassica napus L.). Asian Journal of Agriculture and Biology, 8(4).

Following reference should be cited to add more information about gene family duplication events in brassica.

• Lv, X., Wei, F., Lian, B., Yin, G., Sun, M., Chen, P., ... & Wei, H. (2022). A Comprehensive Analysis of the DUF4228 Gene Family in Gossypium Reveals the Role of GhDUF4228-67 in Salt Tolerance. International Journal of Molecular Sciences, 23(21), 13542.

Reviewer #2: The article entitled "Lineage-specific gene duplication and expansion of DUF1216 gene family in Brassicaceae" is well-compiled manuscript, and the authors performed comprehensive genome-wide analyses on the phylogenetic relationships, gene structures and expressing patterns of DUF1216 members, highlighting its crucial role in flower development of Brassicaceae. In general, the insufficient results are innovative, significant and useful for the research of DUF1216 gene family in Brassicaceae, while many technical issues must be put forward first.

(1)The writing of the manuscript needs improvement, and some grammatical and spelling errors still exist. Suggest to invite someone native to help with the polish.

(2)Normally, numbers should be presented with capitalized word rather than Arabic numerals when being as Subject. Also, Latin names or gene names should be italic, while the authors were not strict with the rules.

(3)The references were not the latest and uniform, as some ones lack the issue number, and some ones use the full names of the cited journal, while some not.

(4)Plenty of gene members were chosen in this study to perform conjoint analyses, while the figures were not big or precise enough to present the results, and suggest to select the specific ones to show in the manuscript.

(5)Only the evolutionary analyses were conducted, while no experimental verification was not supplied in this manuscript.

Reviewer #3: the manuscript reported DUF domain containing gene family in few selected species of family brassicaceae. the manuscript required revision before reaching conclusion. required extensive english editing. moreover, the manuscript lacking functional validation of selected genes.

for instance,

1. the introduction is not related to the topic but random selection of literature. there are plenty of publications reported DUF domain genes author should cite some relative studies here.

2. in last paragraph of introduction, summarize few significant findings as well.

3. line 1226 what are other species? enlist those here.

4. line 133-134 molecular weight should be in KDa.

5. remove discussion from results line 259-287.

6. why expression analysis is done using RNA-seq data from mutants, as in line 393-395.

7. author should characterize few genes for functional validation ?

8. is there any gene with evolutionary importance?

6. PLOS authors have the option to publish the peer review history of their article (what does this mean?). If published, this will include your full peer review and any attached files.

Reviewer #1: No

Reviewer #2: No

Reviewer #3: No

---

## [Author Response · Author response to Decision Letter 0]

17 Mar 2024

Dear Editors：

Thank you for your letter and for the reviewer’s comments concerning our manuscript entitled “Lineage-specific gene duplication and expansion of DUF1216 gene family in Brassicaceae” (ID: PONE-D-24-02626).

We have studied their comments carefully and have made correction which we hope meet with their approval. Revised portion are marked in red in the paper. The main corrections in the paper and the responds to the reviewer’s comments are as following:

Responds to the reviewer’s comments:

Reviewer #1: 

Major comment 1. I suggest authors to check the English by a native English speaker.

Response: Thank you for your advice. We have now worked on both language and readability and have also involved native English speakers for language corrections. We really hope that the language level have been substantially improved.

Major comment 2. Italicize the word “Brassicaceae” througout the manuscript.

Response: Thanks for this comment. We have read this paper carefully and corrected this error in revised manuscript as your comment.

Major comment 3. Avoid starting sentence with a mathematical figure (see line 36).

Response: Thank you for your helpful comments. This error has been corrected in the revision.

Major comment 4. Check for space (see line 36).

Response: We were really sorry for our careless mistakes. Thank you for your reminder. This error has been corrected in the revision.

Major comment 5. All scientific names should be italicized for an example see line 74.

Response: We feel great thanks for your professional review work on our article. This error has been corrected in the revision.

Major comment 6. The section of introduction is underdeveloped. Also, it misses out on several important references like authors gene duplication is the main way to produce new gene in Brassica, such statements need to be cited. The authors must mention about other gene families (at least two) which experienced gene duplication events.

Response: Thanks for this constructive comment. We supplement the references and add two gene families that have undergone expansion in Brassicaceae as examples.

Major comment 7. How do DUF1216 modulate gene duplication?

Response: Thank you for this comment. We have identified a specific expansion of the DUF1216 gene family exclusively in Brassicaceae, marking a significant discovery in our study. The evolutionary history of the DUF1216 gene family has been examined for the first time. Various mechanisms play significant roles in the expansion of DUF1216 genes among different Brassicaceae species. Segmental duplication is the primary mechanism responsible for the expansion of DUF1216 genes in B.oleracea, B.carinata, B.napus, B.juncea, and L.meyneii, while tandem duplication is the main expansion mechanism for the expansion of DUF1216 genes in A.thaliana and C.rubella. As a consequence, a complex interplay of tandem- and segmental duplication drives expansion of DUF1216 gene family in the Brassicaceae.

Major comment 8. To analyze phylogeny, on what basis different listed plants were selected.

Response: Thank you for this comment. When conducting phylogenetic analysis, we employ the following criteria for selecting plant species: (1) Species are chosen based on their representativeness, with a preference for those that receive significant attention and possess high importance in order to facilitate meaningful comparisons. (2) Priority is given to species with well-established genome availability, including those that have undergone complete genome sequencing or possess sufficient reference support, ensuring the availability of comprehensive data and information. (3) The selection also considers an optimal evolutionary distance; excessively close or distant evolutionary relationships among species may compromise the validity of the results. Therefore, we fully considered the selection of Brassicaceae species and selected 27 species, including Arabidopsis thaliana.

Major comment 9. In the methodology section, references are not cited for some context for the protocols used for experimentation by the author. Please cite appropriate references along with the text to acknowledge the contribution of other researchers and to justify the authenticity of the protocols used. For example, cite for HMMER, IQ-TREE.

Response: This is a very important comment for us to improve our manuscript. We revised the manuscript according to your advice. Please find the changes in the manuscripts.

Major comment 10. There is tense inconsistency in methodology. Kindly check and rectify as it a grammatical error which must be rectified for uniform reading of manuscript.

Response: Thank you for your suggestion. Revised in accordance with your suggestions, we have made the necessary changes to the manuscript. Please refer to the revised version for these modifications.

Major comment 11. Authors must add more clarity about the Ka/Ks related results obtained. What can be deduced if a gene family experience purifying selection pressure.

Response: This is a very important comment for us to improve our manuscript. We revised the manuscript according to your advice. Please find the changes in the manuscripts.

Major comment 12. It is suggested that qRT-PCR will add more meaning in your expression experiment work.

Response: We gratefully appreciate for your valuable suggestion. We incorporated the qRT-PCR experiment into the original text and made necessary modifications. The precise details of the modifications are outlined below.

Gene expression analysis qRT-PCR validation

To verify the expression profiles of DUF1216 genes, we selected and analyzed the expression of Arabidopsis DUF1216 genes with qRT-PCR (Fig 9). The expression levels of the tested Arabidopsis DUF1216 genes were specifically high expressed in mature flowers and pollen grains, but not in vegetable tissues (leaf, root, stem) (Fig 9A). These expression patterns are similar to transcriptomic results (Fig 7), indicating that Arabidopsis DUF1216 genes involved in reproductive development. 

Dyt1, TDF1, AMS and MS188 are essential transcription factors that regulate anther development [1-5]. Based on the previous microarray data, most DUF1216 genes act downstream of these regulators (Fig 7) [5]. We collected the inflorescences of dyt1, tdf1, ams, and ms188, for quantitative real-time PCR (qPCR) analysis. The qPCR results showed that the expressions of DUF1216 genes were downregulated in these mutants (Fig 9B). These results demonstrated that Arabidopsis DUF1216 genes were regulated by anther important transcription factors during anther development.

Fig 9. Expression profiles of Arabidopsis DUF1216 genes. A, Expression profiles of Arabidopsis DUF1216 genes in young flower, mature flower and mature pollen. The error bars were obtained from three measurements. B, qRT-PCR analysis of DUF1216 genes in the inflorescence of WT, dyt1, tdf1, ams, and ms188 mutants. Data are presented as mean and SD (n = 3).

Major comment 13. What is the contribution of this research in field of science and to the society? You can add citation from these articles just to add general importance of brassica.

• Farooq, O., Ali, M., Sarwar, N., Mazhar Iqbal, M., Naz, T., Asghar, M., & Ehsan, F. (2021). Foliar applied brassica water extract improves the seedling development of wheat and chickpea. Asian Journal of Agriculture and Biology, 1.

• Ahmad, N., Fazli Ahad, R., Iqbal, T., Khan, N., Nauman, M., & Hameed, F. (2020). Genetic analysis of biochemical traits in F3 populations of rapeseed (Brassica napus L.). Asian Journal of Agriculture and Biology, 8(4).

Response: We gratefully appreciate for your valuable suggestion and providing the excellent reference for us to improve the manuscript. We revised the manuscript according to your advice. Please find the changes in the manuscripts.

Major comment 14. Following reference should be cited to add more information about gene family duplication events in brassica.

• Lv, X., Wei, F., Lian, B., Yin, G., Sun, M., Chen, P., ... & Wei, H. (2022). A Comprehensive Analysis of the DUF4228 Gene Family in Gossypium Reveals the Role of GhDUF4228-67 in Salt Tolerance. International Journal of Molecular Sciences, 23(21), 13542.

Response: We sincerely appreciate your valuable suggestion and the excellent reference you provided to enhance our manuscript. We have made revisions to the manuscript based on your advice. Please review the changes in the revised version.

Reviewer #2: 

The article entitled "Lineage-specific gene duplication and expansion of DUF1216 gene family in Brassicaceae" is well-compiled manuscript, and the authors performed comprehensive genome-wide analyses on the phylogenetic relationships, gene structures and expressing patterns of DUF1216 members, highlighting its crucial role in flower development of Brassicaceae. In general, the insufficient results are innovative, significant and useful for the research of DUF1216 gene family in Brassicaceae, while many technical issues must be put forward first.

Major comment 1. The writing of the manuscript needs improvement, and some grammatical and spelling errors still exist. Suggest to invite someone native to help with the polish.

Response: Thanks for your suggestion. We do invite a friend of us who is a native English speaker from the USA to help polish our article. And we hope the revised manuscript could be acceptable for you.

Major comment 2. Normally, numbers should be presented with capitalized word rather than Arabic numerals when being as Subject. Also, Latin names or gene names should be italic, while the authors were not strict with the rules.

Response: Thank you for your constructive comment. We have read this paper carefully and corrected this error in revised manuscript as your comment.

Major comment 3. The references were not the latest and uniform, as some ones lack the issue number, and some ones use the full names of the cited journal, while some not.

Response: We gratefully appreciate for your valuable suggestion. Based on your comment, we have revised and updated the reference format.

Major comment 4. Plenty of gene members were chosen in this study to perform conjoint analyses, while the figures were not big or precise enough to present the results, and suggest to select the specific ones to show in the manuscript.

Response: This comment is of utmost importance for enhancing the quality of our manuscript. Considering the extensive scope and depth of our study, we deemed it essential to encompass all genes in our analysis to ensure comprehensiveness and objectivity. However, we have made adjustments to the font size and formatting of all the charts in order to optimize their appearance on A4 paper. Additionally, Fig3 was utilized as a supplementary graph from which representative species data were selected and integrated into the main graph. Our primary objective is to present the data in a lucid manner within the article while also optimizing space utilization.

Major comment 5. Only the evolutionary analyses were conducted, while no experimental verification was not supplied in this manuscript.

Response: We express our sincere appreciation for your valuable suggestion. We have incorporated the qRT-PCR experiment into the original text and made necessary modifications accordingly. The precise details of these modifications are outlined below.

Gene expression analysis qRT-PCR validation

To verify the expression profiles of DUF1216 genes, we selected and analyzed the expression of Arabidopsis DUF1216 genes with qRT-PCR (Fig 9). The expression levels of the tested Arabidopsis DUF1216 genes were specifically high expressed in mature flowers and pollen grains, but not in vegetable tissues (leaf, root, stem) (Fig 9A). These expression patterns are similar to transcriptomic results (Fig 7), indicating that Arabidopsis DUF1216 genes involved in reproductive development. 

Dyt1, TDF1, AMS and MS188 are essential transcription factors that regulate anther development [1-5]. Based on the previous microarray data, most DUF1216 genes act downstream of these regulators (Fig 7) [5]. We collected the inflorescences of dyt1, tdf1, ams, and ms188, for quantitative real-time PCR (qPCR) analysis. The qPCR results showed that the expressions of DUF1216 genes were downregulated in these mutants (Fig 9B). These results demonstrated that Arabidopsis DUF1216 genes were regulated by anther important transcription factors during anther development.

Fig 9. Expression profiles of Arabidopsis DUF1216 genes. A, Expression profiles of Arabidopsis DUF1216 genes in young flower, mature flower and mature pollen. The error bars were obtained from three measurements. B, qRT-PCR analysis of DUF1216 genes in the inflorescence of WT, dyt1, tdf1, ams, and ms188 mutants. Data are presented as mean and SD (n = 3).

Reviewer #3: The manuscript reported DUF domain containing gene family in few selected species of family Brassicaceae. the manuscript required revision before reaching conclusion. required extensive English editing. moreover, the manuscript lacking functional validation of selected genes.

Major comment 1. The introduction is not related to the topic but random selection of literature. there are plenty of publications reported DUF domain genes author should cite some relative studies here. 

Response: This is a very important comment for us to improve our manuscript. We have observed that the DUF1216 gene family exhibits duplication and expansion of lineage-specific genes in Brassicaceae, thereby playing a crucial role in reproductive development. Consequently, we proceed with the formulation of the introduction. Revised based on your valuable feedback, we have enhanced the content and refined the logical structure of the introduction.

Major comment 2. In last paragraph of introduction, summarize few significant findings as well. 

Response: We greatly appreciate your invaluable feedback. We have made revisions to the final paragraph of the introduction.

Major comment 3. Line 126 what are other species? enlist those here. 

Response: Thank you for your constructive comment. Other species are non-Brassicaceae species, such as algae, mosses, ferns, monocotyledonous plants, dicotyledonous plants other than Brassicaceae, etc. We found the DUF1216 genes only in Brassicaceae. We revised the manuscript according to your comment. Please find the changes in the manuscripts.

Major comment 4. Line 133-134 molecular weight should be in KDa.

Response: We gratefully appreciate for your valuable suggestion and providing the excellent reference for us to improve the mansucript. We revised the manuscript according to your advice. Please find the changes in the manuscripts.

Major comment 5. Remove discussion from results line 259-287.

Response: Thank you for your suggestion. Revised in accordance with your suggestions, we have made the necessary changes to the manuscript. Please refer to the revised version for these modifications.

Major comment 6. Why expression analysis is done using RNA-seq data from mutants, as in line 393-395?

Response: Thank you for your constructive comment. We utilized RNA-seq data from mutants for the following rationales: 1. Based on tissue expression data, we observed an association between the DUF1216 gene family and reproductive development in Brassicaceae. 2. Previous studies have demonstrated the crucial role of DYT1, TDF1, AMS, and MS188 as transcription factors in anther development [1-4, 6]. Mutations in these genes can impede anther development and result in male sterility. Consequently, we discovered that most AtDUF1216 genes are down-regulated in these mutant anthers.

Major comment 7. Author should characterize few genes for functional validation? 

Response: We sincerely appreciate your valuable suggestion. We have incorporated the qRT-PCR experiment into the original text and made necessary modifications accordingly. The precise details of these modifications are outlined below.

Gene expression analysis qRT-PCR validation

To verify the expression profiles of DUF1216 genes, we selected and analyzed the expression of Arabidopsis DUF1216 genes with qRT-PCR (Fig 9). The expression levels of the tested Arabidopsis DUF1216 genes were specifically high expressed in mature flowers and pollen grains, but not in vegetable tissues (leaf, root, stem) (Fig 9A). These expression patterns are similar to transcriptomic results (Fig 7), indicating that Arabidopsis DUF1216 genes involved in reproductive development. 

Dyt1, TDF1, AMS and MS188 are essential transcription factors that regulate anther development [1-5]. Based on the previous microarray data, most DUF1216 genes act downstream of these regulators (Fig 7) [5]. We collected the inflorescences of dyt1, tdf1, ams, and ms188, for quantitative real-time PCR (qPCR) analysis. The qPCR results showed that the expressions of DUF1216 genes were downregulated in these mutants (Fig 9B). These results demonstrated that Arabidopsis DUF1216 genes were regulated by anther important transcription factors during anther development.

Fig 9. Expression profiles of Arabidopsis DUF1216 genes. A, Expression profiles of Arabidopsis DUF1216 genes in young flower, mature flower and mature pollen. The error bars were obtained from three measurements. B, qRT-PCR analysis of DUF1216 genes in the inflorescence of WT, dyt1, tdf1, ams, and ms188 mutants. Data are presented as mean and SD (n = 3).

Major comment 8. Is there any gene with evolutionary importance? 

Response: The DUF1216 gene family is exclusively present in Brassicaceae and exhibits specific high expression in the anther tissue. Furthermore, these genes are positioned downstream of transcription factors DYT1, TDF1, AMS, and MS188, implying their crucial involvement in anther development under the regulation of transcription factors. These genes hold significant importance for plant reproductive development despite their functions remaining partially elucidated. We are currently conducting further studies to validate gene function and explore the underlying mechanism.

References

1. Sorensen AM, Kröber S, Unte US, Huijser P, Dekker K, Saedler H. The Arabidopsis ABORTED MICROSPORES (AMS) gene encodes a MYC class transcription factor. The Plant journal : for cell and molecular biology. 2003;33(2):413-23. Epub 2003/01/22. doi: 10.1046/j.1365-313x.2003.01644.x. PubMed PMID: 12535353.

2. Zhang W, Sun Y, Timofejeva L, Chen C, Grossniklaus U, Ma H. Regulation of Arabidopsis tapetum development and function by DYSFUNCTIONAL TAPETUM1 (DYT1) encoding a putative bHLH transcription factor. Development (Cambridge, England). 2006;133(16):3085-95. Epub 2006/07/13. doi: 10.1242/dev.02463. PubMed PMID: 16831835.

3. Zhang ZB, Zhu J, Gao JF, Wang C, Li H, Li H, et al. Transcription factor AtMYB103 is required for anther development by regulating tapetum development, callose dissolution and exine formation in Arabidopsis. The Plant journal : for cell and molecular biology. 2007;52(3):528-38. Epub 2007/08/31. doi: 10.1111/j.1365-313X.2007.03254.x. PubMed PMID: 17727613.

4. Zhu J, Chen H, Li H, Gao JF, Jiang H, Wang C, et al. Defective in Tapetal development and function 1 is essential for anther development and tapetal function for microspore maturation in Arabidopsis. The Plant journal : for cell and molecular biology. 2008;55(2):266-77. Epub 2008/04/10. doi: 10.1111/j.1365-313X.2008.03500.x. PubMed PMID: 18397379.

5. Zhu J, Lou Y, Xu X, Yang ZN. A genetic pathway for tapetum development and function in Arabidopsis. Journal of integrative plant biology. 2011;53(11):892-900. Epub 2011/10/01. doi: 10.1111/j.1744-7909.2011.01078.x. PubMed PMID: 21957980.

6. Li DD, Xue JS, Zhu J, Yang ZN. Gene Regulatory Network for Tapetum Development in Arabidopsis thaliana. Frontiers in plant science. 2017;8:1559. Epub 2017/09/29. doi: 10.3389/fpls.2017.01559. PubMed PMID: 28955355; PubMed Central PMCID: PMCPMC5601042.

---

## [Decision Letter · Decision Letter 1]

2 Apr 2024

Lineage-specific gene duplication and expansion of DUF1216 gene family in Brassicaceae

PONE-D-24-02626R1

Dear Dr. Zhang,

We’re pleased to inform you that your manuscript has been judged scientifically suitable for publication and will be formally accepted for publication once it meets all outstanding technical requirements.

Kind regards,

Achraf El Allali, PhD

Academic Editor

PLOS ONE

Additional Editor Comments (optional):

Reviewers' comments:

Reviewer's Responses to Questions

**Comments to the Author**

1. If the authors have adequately addressed your comments raised in a previous round of review and you feel that this manuscript is now acceptable for publication, you may indicate that here to bypass the “Comments to the Author” section, enter your conflict of interest statement in the “Confidential to Editor” section, and submit your "Accept" recommendation.

Reviewer #1: All comments have been addressed

Reviewer #2: All comments have been addressed

Reviewer #3: All comments have been addressed

2. Is the manuscript technically sound, and do the data support the conclusions?

Reviewer #1: Yes

Reviewer #2: Yes

Reviewer #3: Yes

3. Has the statistical analysis been performed appropriately and rigorously? 

Reviewer #1: Yes

Reviewer #2: Yes

Reviewer #3: Yes

4. Have the authors made all data underlying the findings in their manuscript fully available?

Reviewer #1: Yes

Reviewer #2: Yes

Reviewer #3: Yes

5. Is the manuscript presented in an intelligible fashion and written in standard English?

Reviewer #1: Yes

Reviewer #2: Yes

Reviewer #3: Yes

6. Review Comments to the Author

Reviewer #1: (No Response)

Reviewer #2: The article entitled "Lineage-specific gene duplication and expansion of DUF1216 gene family in Brassicaceae" is well-compiled manuscript, and the authors performed comprehensive genome-wide analyses on the phylogenetic relationships, gene structures and expressing patterns of DUF1216 members, highlighting its crucial role in

flower development of Brassicaceae. In general, the insufficient results are innovative, significant and useful for the research of DUF1216 gene family in Brassicaceae, and all the incorrect portions have been revised.

Reviewer #3: (No Response)

7. PLOS authors have the option to publish the peer review history of their article (what does this mean?). If published, this will include your full peer review and any attached files.

Reviewer #1: **Yes: **Shoaib Ur Rehman

Reviewer #2: No

Reviewer #3: **Yes: **MUHAMMAD WASEEM

---

## [Editor Report · Acceptance letter]

4 Apr 2024

PONE-D-24-02626R1 

PLOS ONE

Dear Dr. Zhang, 

I'm pleased to inform you that your manuscript has been deemed suitable for publication in PLOS ONE. Congratulations! Your manuscript is now being handed over to our production team.

Kind regards, 

on behalf of

Dr. Achraf El Allali 

Academic Editor

PLOS ONE